



# Sensitivity of the tropical climate to an interhemispheric thermal gradient: the role of tropical ocean dynamics

Stefanie Talento[1,2], Marcelo Barreiro[1]

[1]Department of Atmospheric Sciences, Institute of Physics, Universidad de la República, Montevideo, 11400, Uruguay

[2]Department of Geography, Climatology, Clim Dynam and Climate Change, Justus Liebig University of Giessen, 35390, Giessen, Germany

*Correspondence to*: Stefanie Talento (stefanie.talento@geogr.uni-giessen.de)

**Abstract.** This study aims to determine the role of the tropical ocean dynamics in the response of the climate to an extratropical thermal forcing. We analyse and compare the outcomes of coupling an atmospheric general circulation model

(AGCM) with two ocean models of different complexity. In the first configuration the AGCM is coupled with a slab ocean model while in the second a Reduced Gravity Ocean (RGO) model is additionally coupled in the tropical region. We find that the imposition of an extratropical thermal forcing (warming in the Northern Hemisphere and cooling in the Southern Hemisphere with zero global mean) produces, in terms of annual means, a weaker response when the RGO is coupled, thus indicating that the tropical ocean dynamics opposes the incoming remote signal. On the other hand, while the slab ocean

coupling does not produce significant changes to the equatorial Pacific sea surface temperature (SST) seasonal cycle, the RGO configuration generates a strong warming in the centre-east of the basin from April to August balanced by a cooling during the rest of the year, strengthening the seasonal cycle in the eastern portion of the basin. We hypothesize that such changes are possible via the dynamical effect that zonal wind stress has on the thermocline depth. We also find that the imposed extratropical pattern affects El Niño Southern Oscillation, weakening its amplitude and low-frequency behaviour.

## 1.        Introduction

Paleoclimatic data (Wang et al., 2004), 20[th] century observations (Folland et al., 1986) and numerical simulations (Chiang and Bitz 2005; Kang et al., 2008, 2009; Cvijanovic and Chiang 2013; Talento and Barreiro 2016, 2017) have all suggested the capability of an extratropical thermal forcing to affect different features of the tropical climate. The general picture emerging from these studies is that the Inter Tropical Convergence Zone (ITCZ) tends to shift toward the warmer

hemisphere at the same time that the atmospheric energy transport is modified to favour the transmission of energy to the colder hemisphere.

In particular, Talento and Barreiro (2016) use an atmospheric general circulation model (AGCM) coupled to a slab ocean model to quantify the relative roles of the atmosphere, tropical sea surface temperatures (SST) and continental surface

temperatures in the ITCZ response to an extratropical thermal forcing. They find that if the tropical SSTs are not allowed to



change, then the ITCZ response strongly weakens although it is not negligible, in particular, over the Atlantic Ocean and Africa. If, in addition, the land surface temperature over Africa is maintained fixed the ITCZ response completely vanishes, indicating that the ITCZ response to the extratropical forcing is not possible just trough purely atmospheric processes, but needs the involvement of either the tropical SST or the continental surface temperatures. With the same model configuration,

Talento and Barreiro (2017) focus on the South Atlantic Convergence Zone (SACZ) and show that, during its peak in austral summer, its response to a warming in the Northern Hemisphere (NH) extratropics and a cooling in the Southern Hemisphere (SH) extratropics consists of a weakening, mostly due to the NH component of the forcing. Both studies showed strong changes in the tropical band where SST, surface winds and precipitation are strongly coupled. Nevertheless, in these studies important ocean dynamics are missing as the slab ocean can only simulate the thermodynamic exchange between the

atmosphere and the ocean.

Chiang et al., (2008) explore the impact of an interhemispheric thermal gradient (ITG) on the tropical Pacific climate through simulations performed with an AGCM coupled to a medium-complexity model: a Reduced Gravity Ocean (RGO) model. They find that when the NH is warmer than the SH the annual mean equatorial zonal SST gradient strengthens,

associated with an earlier onset and a later retreat of the seasonal cold tongue together with an intensification during the peak cold season. They also find that El Niño Southern Oscillation (ENSO) activity is sensitive to the ITG, being small ITG optimal for the development of ENSO activity.

Lee et al., (2015) also use an AGCM coupled to a RGO model and analyse the impact of the glacial continental ice sheet

topography on the tropical Pacific climate. They suggest that the thickness of the ice sheets, separate from the ice albedo effect, has a significant impact on the tropical climate. They identify two types of responses: a quasi-linear response directly associated with the topographic changes and a nonlinear response mediated through the tropical thermocline adjustment. They find that increasing the thickness of the continental ice sheets produces a southward displacement of the ITCZ and a weakening of the equatorial zonal SST gradient, caused by cooling (warming) in the western (eastern) equatorial Pacific,

together with a thermocline deepening to the east. They note that the energy flux approach proposed in Kang et al., (2008, 2009) and Cvijanovic and Chiang (2013) does not appear to explain the ITCZ shifts in these experiments because even though the northern cross equatorial energy transport increases with the ice thickness, the mid-latitude transport decreases.

Studies that analyse the extratropical to tropical teleconnection in a hierarchy of model configurations are scarce, as most of

the literature on the subject focuses on just one ocean model at a time. One exception is the work by Kay et al., (2016) in which the authors study the effect of a Southern Ocean cooling on the tropical precipitation, coupling an AGCM either to a slab or to a full oceanic model. They find that with dynamic ocean heat transport the tropical precipitation response is weaker being, in this case, most of the cross-equatorial heat transport carried out by the ocean and not by the atmosphere. Similar

conclusions are obtained by Hawcroft et al., (2016) with a different fully coupled model, suggesting that the results are not

model specific.

In this study we propose to further examine the role of equatorial ocean dynamics in the tropical response to an extratropical forcing in a hierarchy of model configurations. To isolate the tropical ocean dynamics' effect we will analyse and compare the response generated when an AGCM is coupled to a slab ocean model to that obtained when a RGO model is additionally

coupled in the tropical oceans.

The paper is organized as follows. In section 2 we describe the models used, with special emphasis in the description of the RGO model and its validation against observational data. The experiments performed are explained in section 3. The results can be find in section 4, discriminated as regarding to changes in annual mean, seasonal cycle or ENSO. A summary and

conclusions are presented in section 5.

## 2.     Model Description

The atmospheric model used in this study is the Abdus Salam International Centre for Theoretical Physics (ICTP) AGCM (Molteni 2003; Kucharski et al., 2006) which is a full atmospheric model with simplified physics. We use the model version

40 in its 8-layer configuration and T30 (3.75ºx3.75º) horizontal resolution. Present-day boundary surface conditions, orbital parameters and greenhouse forcing are used.

We analyze the outcomes of coupling the AGCM with two ocean models of different complexity. In the first configuration the AGCM is coupled with a slab ocean model; a monthly-varying ocean heat flux correction is imposed in order to keep the

simulated SST close to present-day conditions. In the second configuration, and in order to better reproduce the tropical ocean dynamics, a RGO model is coupled in the tropical region (30ºS-30ºN), while a slab ocean model is applied elsewhere. In this setup an annual-mean ocean heat flux correction is imposed in order to keep the simulated SST close to present-day conditions.

We proceed to describe the RGO model and to validate its results comparing with observational analogous.



## 2.1      RGO Model Formulation and Validation

We use an extension of the classical 1 ½ layer RGO model, introduced by Cane (1979) to study the ENSO phenomenon. The extension of the model, as in Chang (1994), includes thermodynamics of the upper ocean and allows the prediction of the SST.


The model consists of a 50 m depth upper layer in which mass, heat and momentum obey the conservation laws and a lower layer of infinite depth in which the velocity must be null so that the kinetic energy is finite. The approximation is reasonable for the tropical ocean because of the existence of a sharp thermocline which inhibits the downward propagation of waves

generated in the upper ocean (Zebiak, 1985). To better predict changes of the SST a linear and homogeneous frictional layer (assumed to concentrate most of the induced Ekman transport) is added to the model. The subsurface temperature is parameterized in terms of the thermocline depth, the observed annual mean temperature at 50 m depth (from Levitus 1982) and the annual mean thermocline depth when the model is forced by observed wind stress.

The resolution of the RGO model is 1º in latitude and 2º in longitude, applied in the 30ºS-30ºN tropical band. The model is run using an anomaly coupling strategy. That means that, for momentum and heat fluxes, the oceanic and atmospheric components of the model exchange anomalies computed relative to their own model annual mean. The modeled anomalies are then superimposed to the observed annual mean. Sponge layers of 5º wide are introduced at the northern and southern boundaries to eliminate artificial coastal Kelvin waves.


A 70-years Control simulation in which the AGCM is coupled to the RGO in the tropics and to the slab ocean model elsewhere is produced. The last 50 years of the Control run are used for averaging and comparison with observational analogous. We use the NOAA Extended Reconstructed SST V3b (Smith et al., 2008) and the near-surface winds from the NCEP-NCAR Reanalysis (Kalnay et al., 2006), for the period 1979-2013.


With the imposed heat flux correction, in the Control the simulated annual mean SST strongly resembles the observed pattern (not shown). In addition, the model reasonably captures the main characteristics of the seasonal cycle in the equatorial (2ºS-2ºN) Pacific and Atlantic oceans (Figure 1), although in the equatorial Pacific the de-meaned (annual mean removed) simulated SST seasonal cycle is weaker than in the observations. Also, the Pacific cold tongue is not well

developed during SH summer.

The Control simulation also reproduces the main mode of variability in the tropical Pacific Ocean quite realistically both in the spatial and temporal domains (Figure 2). The first coupled pattern arising from a Singular Value Decomposition (SVD) of the monthly SST and surface wind characterizes ENSO and explains 81% (62%) of the variability in the observations





(simulation). The simulated pattern is weaker than the observed and with the SST anomaly maximum located too far

eastward. The phase-locking to the seasonal cycle of the simulated pattern peaks during the end of the calendar year as do in

the observations, but its distribution is more uniform throughout the year. Both simulated and observed spectra show

significant peaks relative to a red noise null hypothesis from 16 to 60 months.

## 2.2      Experimental Design

For each model configuration two runs are produced: a Control run (in which no forcing is applied) and a forced run (in

which an extratropical forcing is imposed). The applied forcing pattern consists in cooling in one hemisphere and warming

in the other poleward of 40°, applied only over ocean grid points, and with a resulting global average forcing equal to zero.

This pattern is similar to the one used in Kang et al., (2008) and in Talento and Barreiro (2016) and it is intended to represent

the asymmetric temperature changes associated with glacial-interglacial and millennial-scale climate variability as well as

the .asymmetric SST pattern characteristic of the global warming trend. The forcing pattern is superposed to a background

state and is obtained as explained in Talento and Barreiro (2016).

The forcing pattern is shown in Figure 3, in which sign convention is positive out of sea and, therefore, positive values of the

forcing could be thought as representing a situation where the atmosphere is dry and colder than the ocean below it so that

there is a strong ocean-to-atmosphere net heat flux. This forcing generates a near surface temperature (NSAT) anomaly

response of up to 16ºC (-20ºC) over the north Atlantic Ocean at 70ºN (over the Ross Sea), as shown in Figure 4. For

comparison, in a climate simulation of the last 21.000 years (TRACE2k experiment, He 2011) anomalies of about -10ºC

(6ºC) are obtained over the North Atlantic (Antarctica) during Heinrich Stadial 1, 18.000 to 15.000 years ago.


As mentioned before we use two ocean models. When the AGCM is coupled to a slab ocean model, the experiments are

named: *Control_slab* and *Forced_slab*. If an RGO is used in the the tropical band while the slab ocean model is applied

elsewhere, the corresponding experiments are named: *Control_slab+rgo* and *Forced_slab+rgo*. In all the simulations the

model was run for 70 years and the last 50 are used for averaging. Running the simulations for 70 years proved to be more

than enough to reach the equilibrium; a time scale of 10 years was estimated to be the time span necessary for adjustment. In

Table 1 we summarize the experiments.

## 3.      Results

First we analyse and compare the annual mean anomalies generated by the extratropical forcing with the two configurations

implemented. Second, we will focus in the tropical Pacific climate and study the changes produced to the seasonal cycle for



both setups. Finally, we will briefly investigate possible changes in ENSO activity when the RGO is coupled in the tropical oceans.

### 3.1 Annual means

In this subsection we compare the results obtained with the two implemented configurations in terms of annual means of different fields. The results are presented in the form of anomalies with respect to the corresponding Control case.

Figure 4 shows the near surface air temperature (NSAT) changes with respect to the corresponding Control for the two configurations. In both experiments there is a generalized warming (cooling) in the NH (SH), while in the southern tropics a

strengthening of the zonal gradient is evident. The most pronounced differences between the two configurations are seen in the tropical region, in which the slab+rgo configuration anomalies tend to be up to 1ºC weaker than the slab configuration anomalies. In particular, the equatorial Pacific cooling seen in the slab configuration is no longer present in the rgo+slab configuration, and the southeastern ocean basins do not cool as much. This suggests that, overall, tropical ocean dynamics tends to oppose changes in the annual mean conditions. In the extratropics the differences between the two configurations are

almost not noticeable, although regions of up to 2ºC are noted in the vicinity of the Antarctic Peninsula and Greenland.

As a consequence, tropical changes in precipitation are weaker when using the RGO: while in both experiments the most pronounced feature is a northward shift of the ITCZ, anomalies for the slab+rgo configuration are much weaker (Figure 5). Also, in the slab configuration strong changes of tropical precipitation are found equally over the three ocean basins, but for

the slab+rgo setup the most intense anomalies are seen over the Atlantic Ocean concurrent with a still relatively strong cross-equatorial SST gradient and suggesting a larger role for continental temperatures in controlling the position of the ITCZ (as in Talento and Barreiro, 2016). Changes in the subtropical convergence zones are also weaker in the slab+rgo configuration, differently from the southward shift seen in the slab configuration. In particular for the case of the South Atlantic Convergence Zone, this result is consistent with Talento and Barreiro (2016), which showed that during southern summer the

weakening of the SACZ is related to the development of a region of strong rainfall in the tropical north Atlantic.

As expected from the above results, both experiments present similar patterns of near surface (950 hPa) wind anomalies (Figure 6): anomalous northward winds associated with the ITCZ northward displacement in the tropics, and anomalous westerly winds over the Southern Ocean. The *Forced_slab+rgo* response is weaker in the tropics but stronger over the

Southern Ocean, compared to the *Forced_slab* response. Similar pictures of a weaker response in the case of slab+rgo configuration can be seen in upper-level winds, mean sea level pressure and mass stream-function (not shown).



To summarize, in Figure 7 we present the northward atmospheric energy transport for the Control and forced runs in the two configurations implemented. As can be seen, while the Control runs display almost an identical transport, the forced runs

significantly disagree in magnitude in the tropical region, with the slab configuration producing the strongest changes with an increase of the transport toward the southern high latitudes. In the perturbed runs, the energy flux equator is located around 17ºN (13ºN) for the slab (slab+RGO) configuration.

**3.2        Seasonal Cycle**


As the previous subsection showed, the most pronounced differences between the two implemented configurations are found in the tropical band. Therefore, for the analysis of variations in the seasonal cycle we will focus on the 30°S-30°N region.

Three-month means of SST and near-surface wind changes for the tropics are shown in Figure 8. In the Pacific Ocean, for

the *Forced_slab* experiment negative SST anomalies are seen reaching the Equator (or even more to the north) in all 4 seasons, being September-November (SON) the period of strongest cooling and the June-August (JJA) period the one in which this negative anomalies have the weakest penetration onto the NH. Meanwhile, for the *Forced_slab+rgo* experiment the negative SST anomalies barely reach the Equator and, in fact, positive anomalies are the ones penetrating onto the SH for the seasons March-May (MAM) and JJA. Consistent with these changes in response, the equatorial anomalous winds in the

slab configuration are mainly easterlies throughout the year, while in the slab+rgo configuration they have a marked northward component and are eastward during MAM season. Also, the equatorial Atlantic tends to warm up during most of the year when using the RGO model.

Equatorial (2ºS-2ºN) de-meaned seasonal cycles for near-surface zonal wind and SST anomalies in the Pacific basin for the

two experimental configurations are shown in Figures 9 and 10, respectively. In *Forced_slab* there are eastward (westward) near-surface wind anomalies from December to May (June to November) distributed along the basin. The positive wind anomalies in the *Forced_slab* are quite uniform along the basin, although there are maximums in the western and eastern ends during December-February (DJF). The negative anomalies during the second half of the year are maximal in the central-eastern basin. In the slab configuration, the equatorial SST response to these wind anomalies is, however, very weak

(Figure 10a). On the other hand, when the RGO is coupled (and although the annual mean anomalies were even weaker than for the slab configuration, Figure 4) there are significant changes occurring to the seasonal cycle of SST in the center-east of the basin: from April to August (October to December) the forced run produces a warming (cooling) of up to 1ºC (-0.8ºC). The location and timing of these anomalies lead to a substantial strengthening of the SST seasonal cycle in the eastern Pacific Ocean (overlap Figure 10b on the *Control_slab+rgo* SST seasonal cycle showed in Figure 1c).


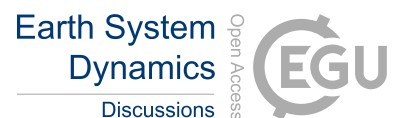

The thermocline depth shows consistent changes when the RGO is used (Figure 11): a deepening in the east of the basin starting around March and finishing in July, consistent with the warmer SSTs seen in the region (with 1 month lag). Considering the wind anomalies of the slab setup (Figure 9a) as the forcing pattern for the ocean dynamics derived from the extratropical signal, this thermocline deepening pulse appears to be initiated during the SH summer in the west of the basin
(due to a weakening of the trades) and is propagated eastward as a Kelvin wave, reaching the eastern boundary 2-3 months later. The deepening of the eastern Pacific thermocline is concurrent with a shallowing in the western Pacific particularly from May to July, and vice versa (but less obvious) in other seasons of the year. In the second half of the year the strengthening of the trades locally shallows the thermocline in the eastern Pacific and the western Pacific recovers its mean depth.


In summary, the equatorial near-surface zonal wind changes caused by the extratropical forcing seen in the slab configuration induce dynamical ocean-atmosphere coupling that generates seasonal changes in the SST field when the RGO is used. This results in a late austral fall warming and a cooling in spring and summer in the equatorial eastern Pacific, leading to a strengthening of the SST seasonal cycle with consistent changes in the thermocline depth.


### 3.3 ENSO

In this subsection we investigate how the interannual variability in the tropical Pacific is affected by the interhemispheric SST gradient induced by the imposed forcing.

The leading pattern of co-variability of SST and near-surface wind in the tropical Pacific basin when the extratropical forcing is applied is weaker than that obtained when no forcing is implemented (Figure 12 and Figure 1 a ), and explains a smaller percentage of the total variability (46% compared to 62%). The phase-locking to the seasonal cycle (Figure 12 b and Figure 1 c) is also modified being more uniformly distributed and with a peak season going from July to the end of the calendar year. The frequency spectrum of the ENSO pattern under the effect of the extratropical forcing is characterized by
shorter periods than in the absence of the forcing and has a peak at 24 months (Figure 12 c).

The weakening of the ENSO activity can be understood in relation to the changes produced by the extratropical forcing on the SST seasonal cycle in the eastern Pacific Ocean. According to the non-linear frequency entrainment mechanism (Chang et al.,, 1994) ENSO amplitude is anticorrelated with the strength of the SST seasonal cycle. The frequency entrainment
implies that a self exciting oscillator (like ENSO) will give up its intrinsic mode of oscillation in the presence of a strong external forcing (like a strong seasonal SST cycle) and acquire the frequency of the applied oscillating forcing. Therefore, in our case, as the extratropical forcing generates a significant strengthening of the east Pacific SST seasonal cycle, a weakening of ENSO is expected according to this mechanism.





Assuming a linear behaviour holds, our result of ENSO weakening is also in agreement with Timmermann et al., (2007). These authors analyse fully coupled GCMs in the context of an Atlantic Meridional Overturning Circulation (AMOC) slowdown, producing a generalized cooling of the NH and warming in the SH, and find that most of the models predict a ENSO intensification attributed to a seasonal cycle weakening. The weakening of ENSO activity in the presence of an northward ITG is also consistent with the work of Chiang et al., (2008) who use a model configuration similar to ours: an

AGCM coupled to a RGO model. Although they do not attempt to explain the causes, they find that ENSO is sensitive to ITG with maximal activity when the ITG is close to zero and a weakened performance as the gradient increases in any direction.

## 4.        Summary and Conclusions


We investigated and compared the response of the tropical climate to an extratropical thermal forcing in a hierarchy of models in which an AGCM was coupled either to a simple slab ocean model (just thermodynamic coupling) globally or with a combination of a RGO model in the tropical oceans and a slab ocean model elsewhere.

First, we found that the responses produced by the two types of configurations greatly differ in the tropical regions, being the signal produced with the RGO coupling weaker in terms of annual means, indicating that regional dynamical air-sea interaction opposes to the remote signal. This result is in agreement with Kay et al., (2016) who also obtained a weaker tropical response to an extratropical thermal forcing when using a fully coupled model than when the AGCM is only coupled to a slab model. However, although the annual mean anomalies produced by the RGO setup are weaker, the changes in the

SST seasonal cycle are larger. In particular, over the equatorial Pacific Ocean, while the slab configuration produces no changes to the SST seasonal cycle, the RGO addition generates a profound warming in the centre-east of the basin from April to August balanced by a cooling in the rest of the year, yielding an almost null integration in the annual mean but also implying a significant strengthening of the seasonal cycle in the eastern Pacific. The response of the seasonal cycle to the imposed extratropical forcing is qualitatively similar to the one obtained by Chiang et al., (2008) in similar experiments,

although in our case positive SST anomalies reach the eastern boundary of the basin not permitting an earlier onset of the seasonal cold tongue as these authors find in their simulations. We hypothesize that the changes in the SST seasonal cycle are possible via the effect that the zonal wind stress has on the thermocline depth: the remote forcing produces positive anomalies of zonal wind stress to be exerted in the first half of the calendar year; in particular, the significant weakening of the trades over the western portion of the basin around February and March induces a thermocline deepening pulse that

propagates eastward in the form of a Kelvin wave, reaching the eastern boundary 2 months later, and generating a warming

of the SST over that region as a result. In the second half of the year stronger trades in the central-eastern basin shallow the thermocline producing a local cooling of the SST. Since these mechanisms are not available under the slab configuration, the wind stress seasonal cycle changes are not able to produce any SST changes.

Finally, we briefly analysed within the RGO setup, possible changes in ENSO activity and found that under the effect of the extratropical forcing significant changes are produced both in the spatial and temporal domains with a weaker SST pattern and a time series that lacks low-frequency variability. We hypothesized that the weakening of the ENSO activity concurrent with the intensification of the SST seasonal cycle in the eastern equatorial Pacific Ocean, could be due to the frequency entrainment mechanism. As future climate projections tend to agree in the fact that global warming will have an important

northward ITG component (NH warming faster than the SH; Friedman et al.,, 2013), the possible sensitivity of ENSO to ITG is of primary relevance. However, current state of the art fully coupled climate models do not seem to agree in the projected future changes in ENSO characteristics and no clear evidence for a correlation with ITG has been detected in future climate projections (Stevenson, 2012; Tascheto et al.,, 2014).

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



Table 1: Experiment summary.

| Experiment Name | Ocean model | Forcing pattern H |
|---|---|---|
| *Control_slab* | Slab ocean model globally | No forcing |
| *Forced_slab* | Slab ocean model globally | Extratropical forcing, as in Figure 3 |
| *Control_slab+rgo* | RGO model in 30°S-30°N, slab ocean model elsewhere | No forcing |
| *Forced_slab+rgo* | RGO model in 30°S-30°N, slab ocean model elsewhere | Extratropical forcing as in Figure 3 |






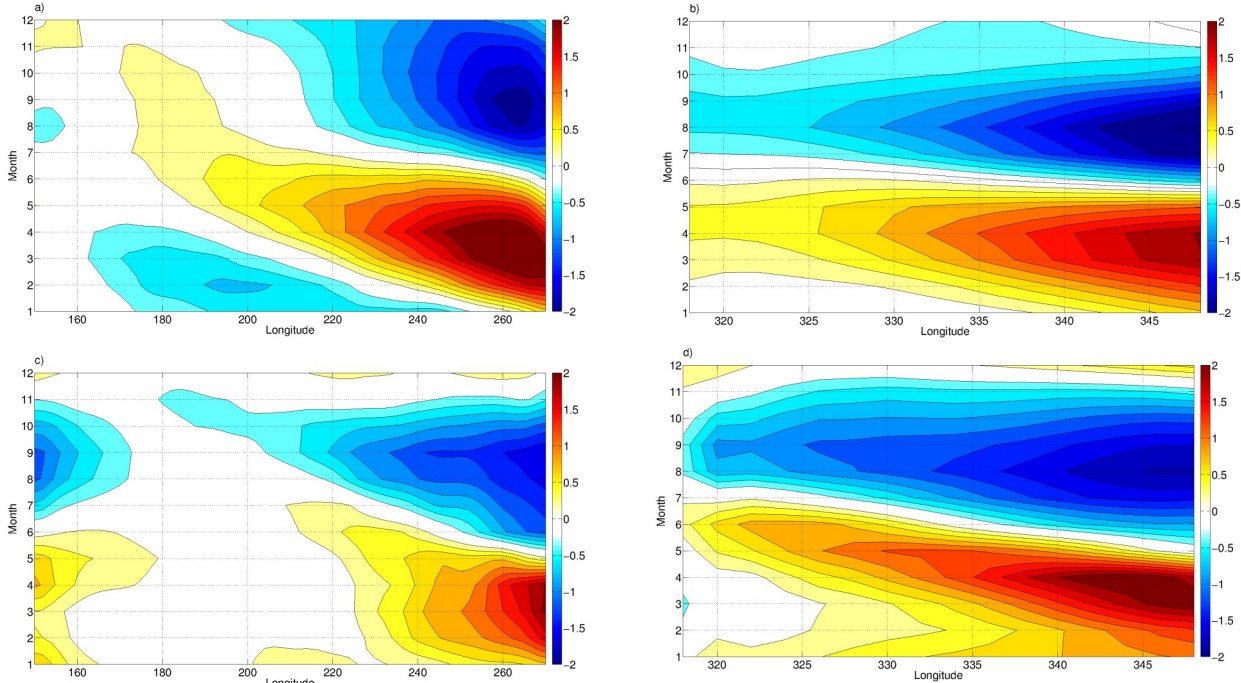

**Figure 1: a. and b.: De-meaned SST seasonal cycle from NOAA SST data (Smith et al.,, 2008) in the Equatorial Pacific (2°S- 2°N, 150°E-270°E) and Atlantic (2°S-2°N, 320°E-345°E), respectively. c. and d.: De-meaned SST seasonal cycle for *Control_slab+rgo* in the Equatorial Pacific and Atlantic, respectively. Contour interval: 0.2°C.**

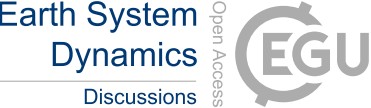




**Figure 2: First SVD pattern of SST and near-surface winds in the tropical Pacific Ocean (30°S-30°N, 120°E-300°E) for the *Control_slab+rgo* experiment (left) and NOAA SST and reanalysis data (right; Smith et al.,, 2008; Kalnay et al.,, 2006). a. and b.: Spatial pattern; contour interval 0.2°C. c. and d.: Histogram showing phase- locking to the seasonal cycle. e. and f.: Spectral analysis, the red line indicates the red noise spectrum.**




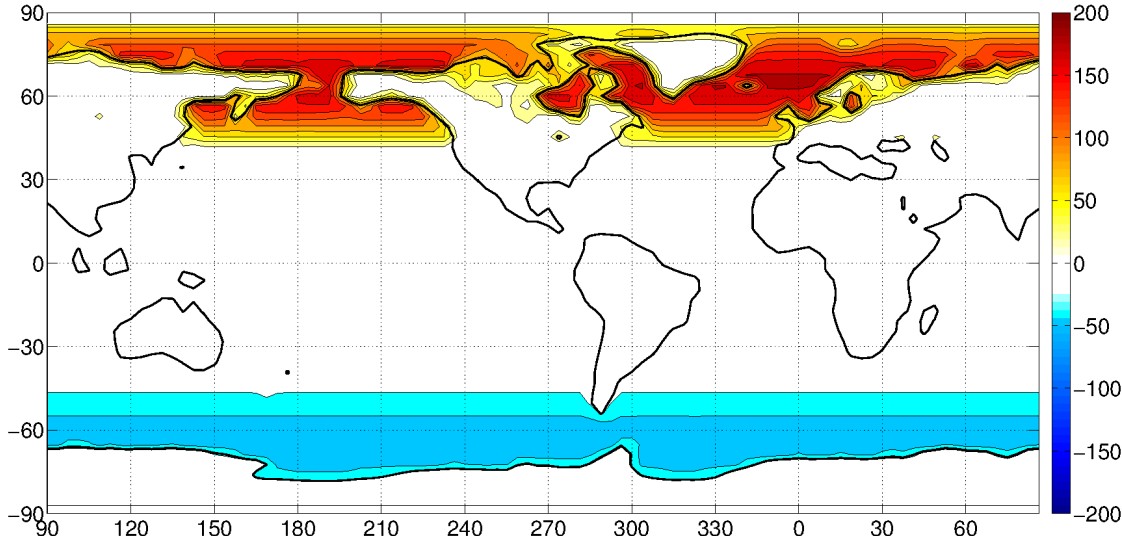

**Figure 3: Forcing pattern. The sign convention is positive out of sea. Contour interval 20 W/m².**





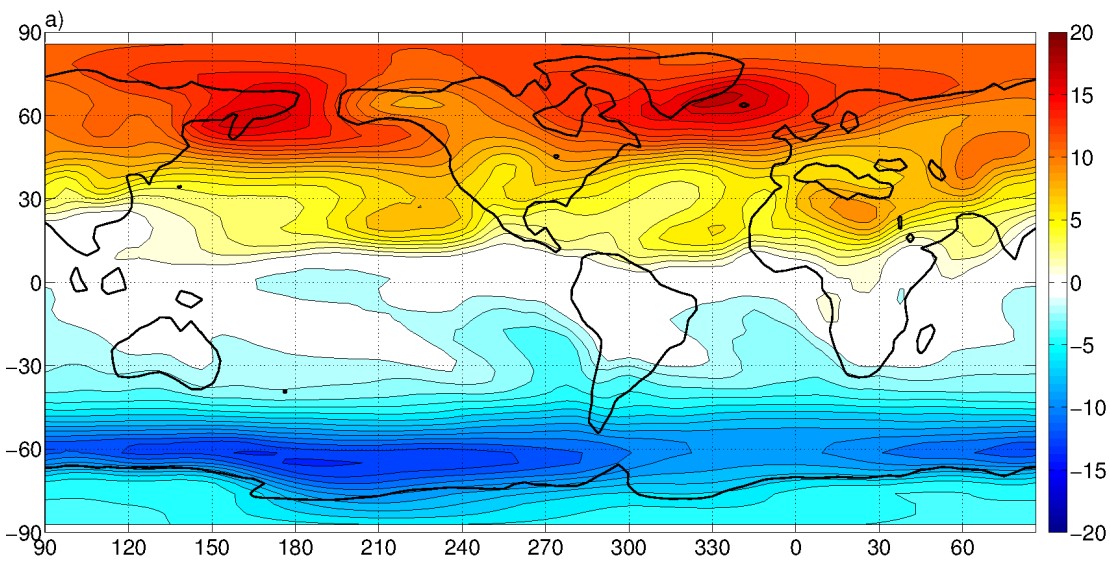

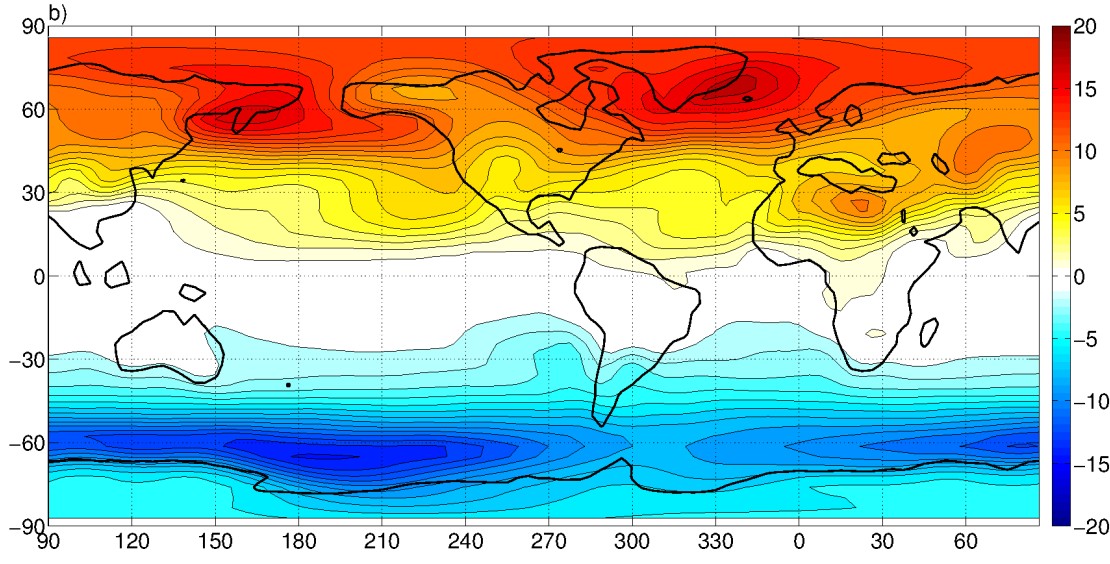

**Figure 4: Annual mean anomalies with respect to the control of NSAT for: a.** *Forced_slab* **and b.** *Forced_slab+rgo*, **respectively. Contour interval 1°C.**





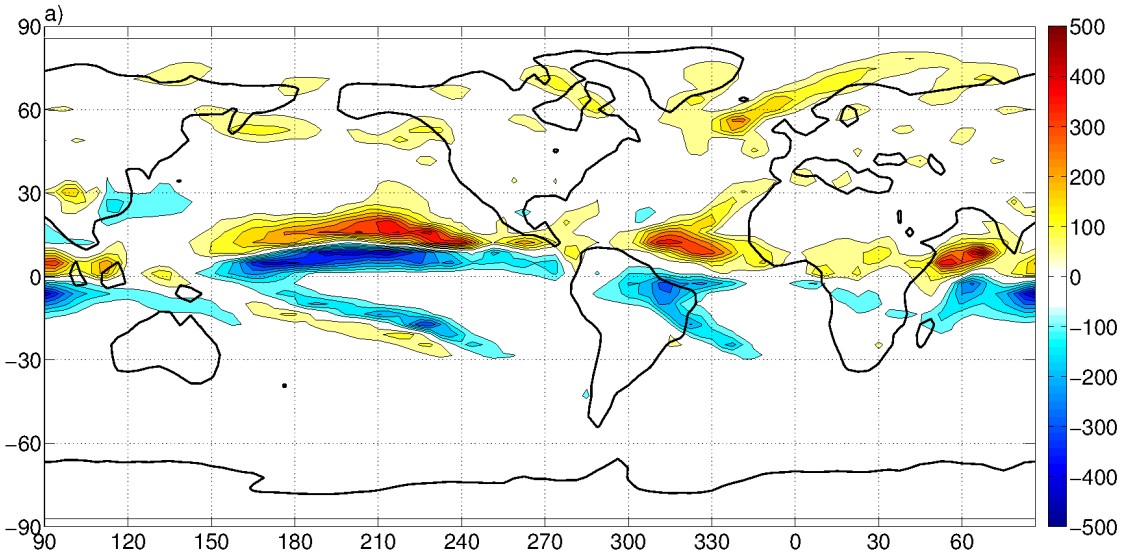

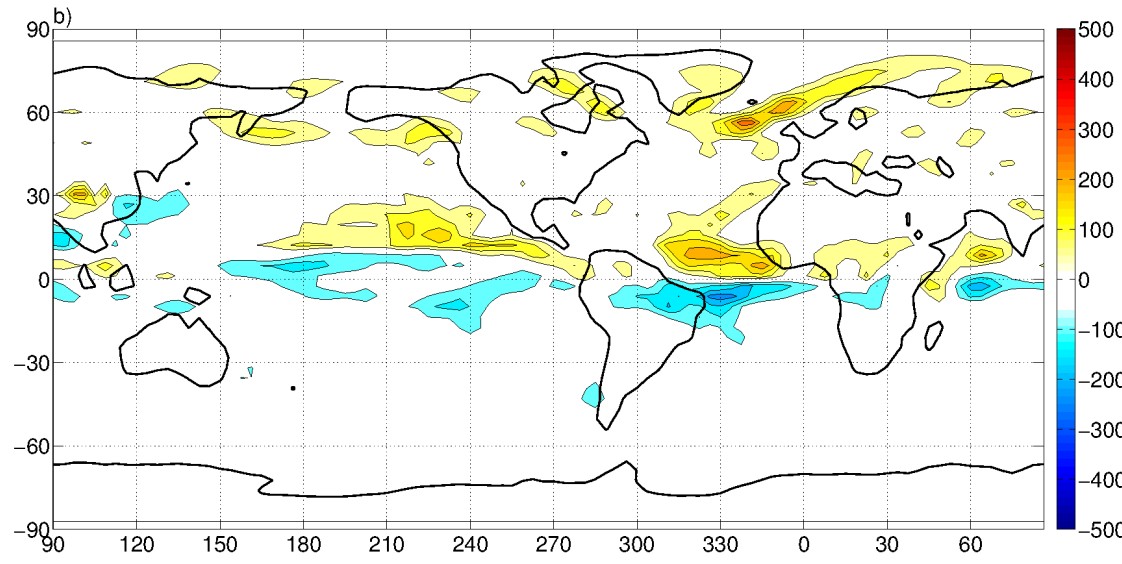

**Figure 5: Annual mean anomalies with respect to the control of precipitation for: a. *Forced_slab* and b. *Forced_slab+rgo*, respectively. Contour interval 50 mm/month.**



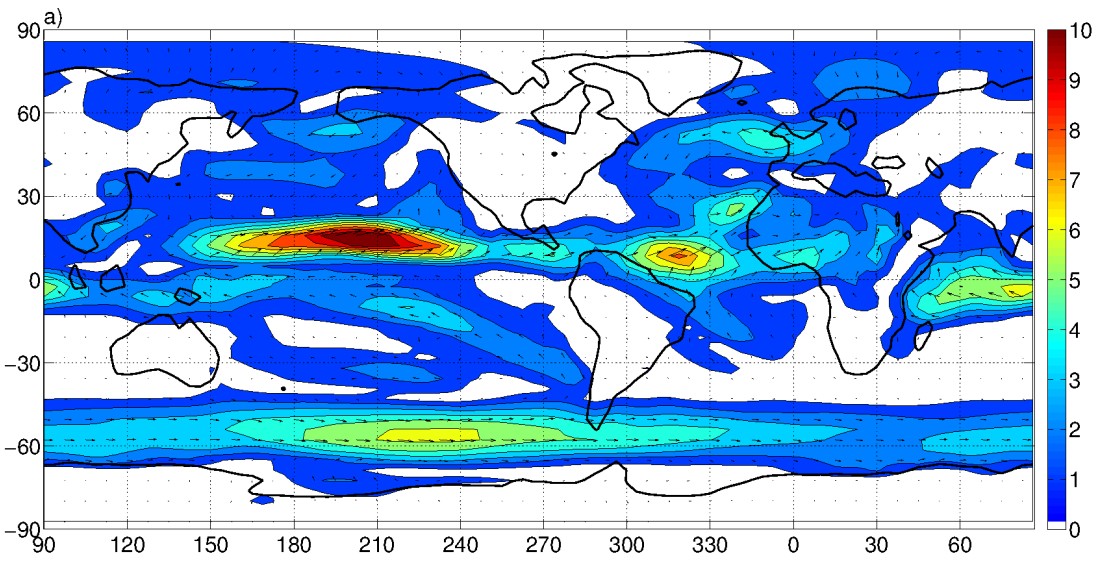

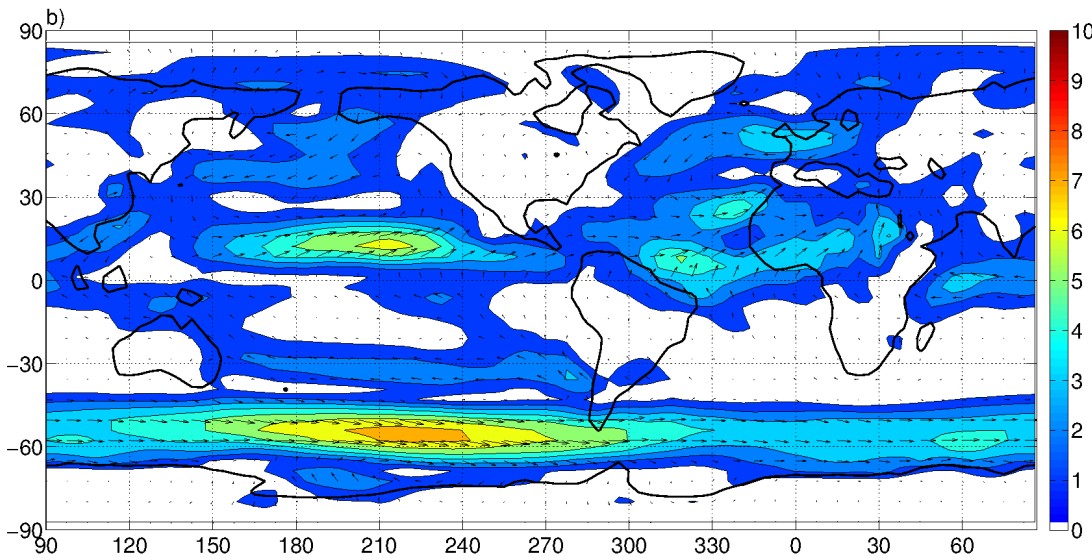

**Figure 6: Annual mean anomalies with respect to the control of near-surface (950 hPa) wind for: a. *Forced_slab* and b. *Forced_slab+rgo*, respectively. Contour interval 1 m/s.**






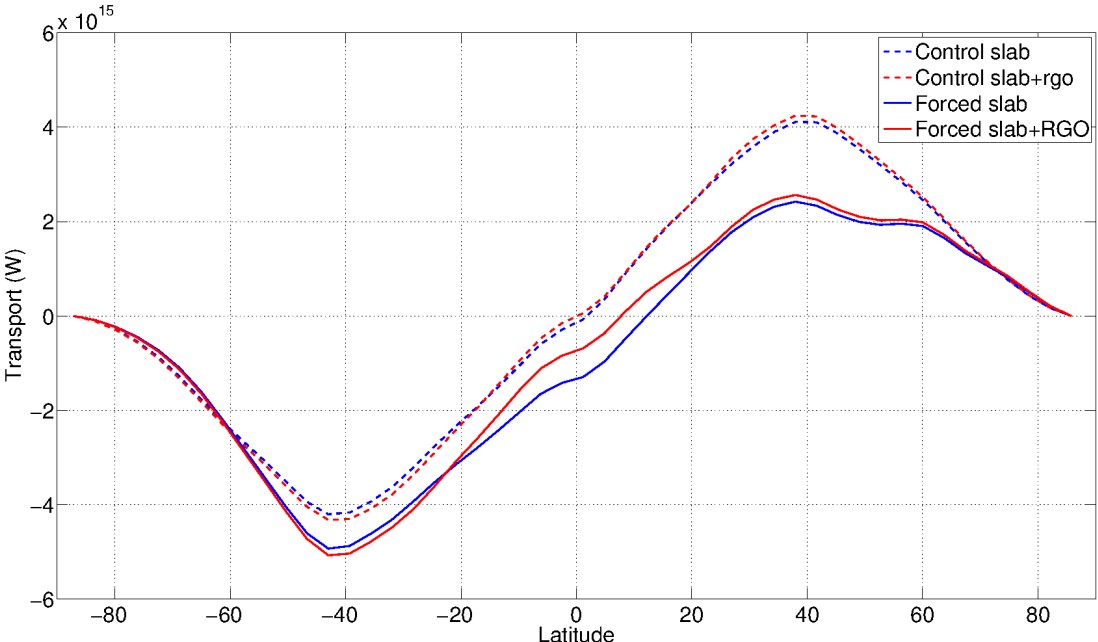

**Figure 7: Northward atmospheric energy transport for the experiments:** *Control_slab*, *Control_slab+rgo*, *Forced_slab* **and** *Forced_slab+rgo.*



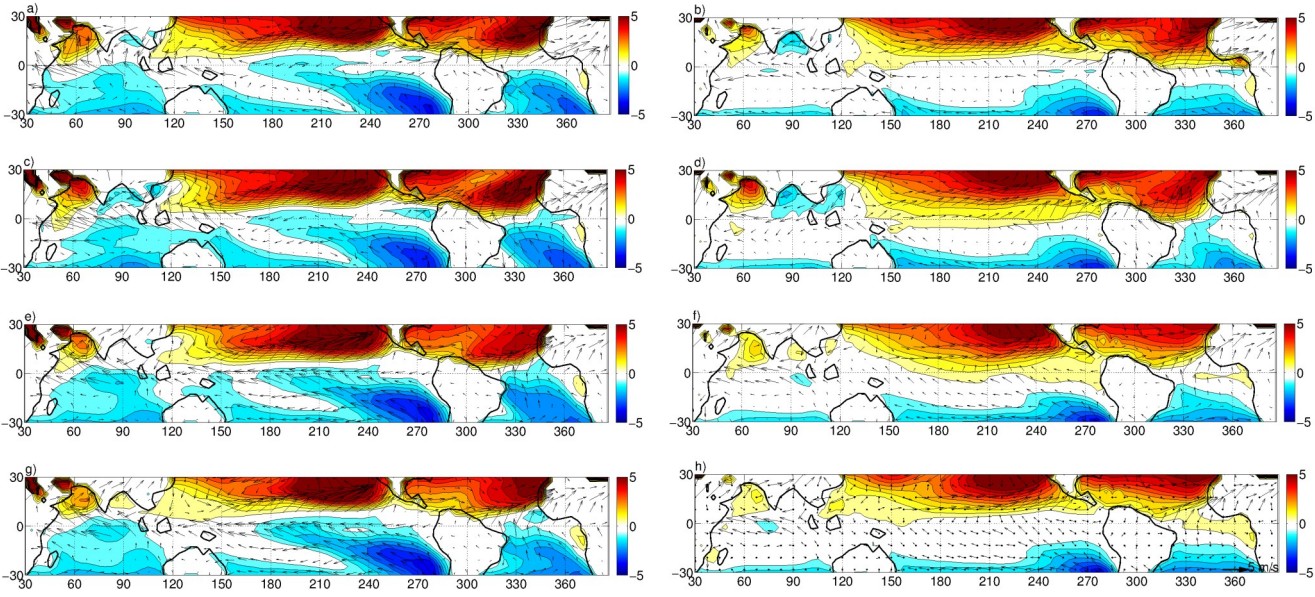


**Figure 8: Seasonal SST and near-surface wind anomalies with respect to the control for Forced_slab (left) and Forced_slab+rgo (right). a. and b.: December-February; c. and d. March-May; e. and f.: June-August; g. and h.: September-November. Contour interval 0.5°C.**

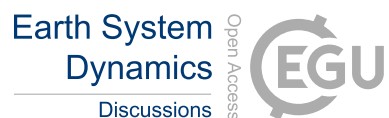

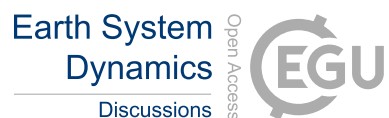


**Figure 9: Equatorial Pacific Ocean (2°S-2°N, 150°E-270°E) de-meaned near-surface (950 hPa) zonal wind anomalies seasonal cycle for: a.** *Forced_slab* **and b.** *Forced_slab+rgo* **experiments, respectively. Contour interval: 0.5 m/s.**





**Figure 10: Equatorial Pacific Ocean (2°S-2°N, 150°E-270°E) de-meaned SST anomalies seasonal cycle for: a.** *Forced_slab* **and b.** *Forced_slab+rgo* **experiments, respectively. Contour interval: 0.2°C.**





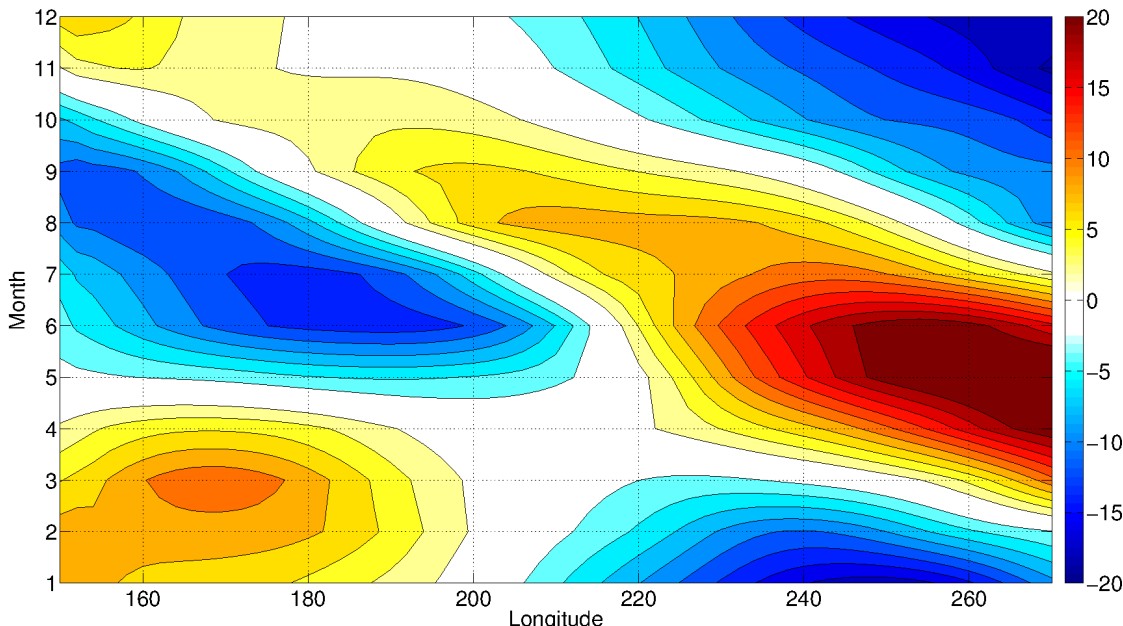

**Figure 11: Equatorial Pacific Ocean (2°S-2°N, 150°E-270°E) de-meaned thermocline depth anomalies seasonal cycle**
**for: a. *Forced_slab* and b. *Forced_slab+rgo* experiments, respectively. Contour interval: 2 cm**



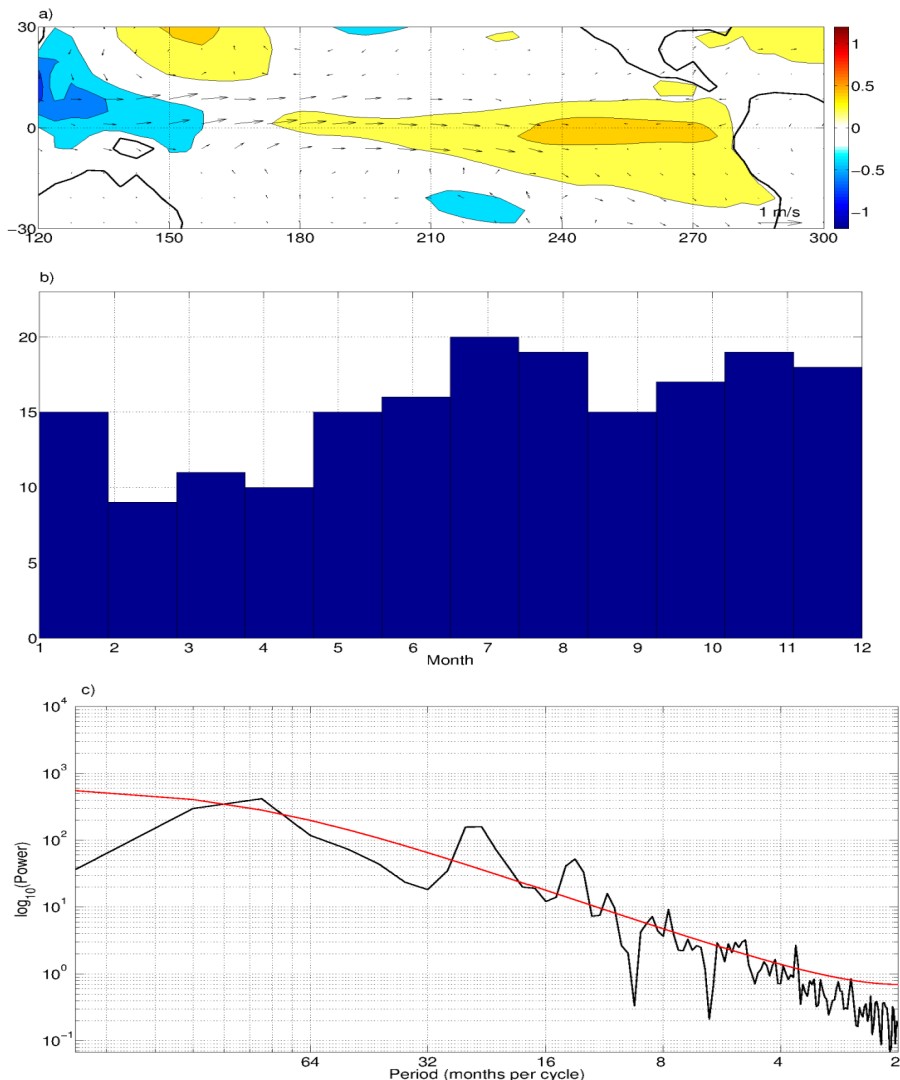

**Figure 12: First SVD pattern of SST and near-surface winds in the tropical Pacific Ocean (30°S-30°N, 120°E-300°E)**
**for the *Forced_slab+rgo* experiments. a.: Spatial pattern; contour interval 0.2°C. b.: Histogram showing phase-locking to the seasonal cycle. c.: Spectral analysis; the red line indicates the red noise spectrum.**