# Peer review of "Sensitivity of the tropical climate to an interhemispheric thermal gradient: the role of tropical ocean dynamics"

_Earth System Dynamics, 2017_

## Referee Comment (RC1) · Anonymous Referee #1 · 24 Dec 2017

General Comments

The main motivation of this manuscript titled "Sensitivity of the tropical climate to an interhemispheric thermal gradient: the role of tropical ocean dynamics" is to understand how the addition of a simple dynamic ocean model in the tropics affects the climate response to idealized extratropical forcing with different signs in each hemisphere. Four coupled simulations are presented, two that are coupled to a slab ocean model (SOM) and two that are coupled to a reduced gravity ocean (RGO) model. The extratropical forcing is applied to one simulation in each of these model pairs. The main results are that the tropical precipitation shift in response to the forcing is weaker in the model with

the RGO and that the seasonal cycle in the tropics has large change in response to the forcing in the model with the RGO but not the SOM.

To my best knowledge, the examination of ITCZ shifts to extratropical thermal forcing with a climate model coupled to a reduced gravity ocean has not been done yet. This examination is timely given the results presented in Kay et al. (2016) and Hawcroft et al. (2017) that show a clear role for ocean dynamics in affecting tropical precipitation shifts. That said, this manuscript misses a few key studies from the past year that are very relevant: Green and Marshall (2017) and Schneider (2017). These two studies provide a mechanistic explanation for how tropical ocean circulation damps an ITCZ shift through coupling of the ocean and atmosphere by wind stress (see comments below). The results in this manuscript support the arguments in these two papers, and I believe that further discussion in light of Green's and Schneider's arguments would strengthen this manuscript. Otherwise, I find this manuscript clearly argued and written, and I only have a few other minor comments, mostly regarding figure quality and additional related literature. I recommend this paper for acceptance pending a major revision to include discussion of Green and Schneider and the minor issues listed below.

Specific Comments

Major comment 1: Green and Marshall (2017) (DOI: 10.1175/JCLI-D-16-0818.1) and Schneider (2017) (DOI: 10.1002/2017GL075817) have both shown that an ITCZ shift is damped by tropical ocean circulation, specifically by meridional heat transport through Ekman transport by the subtropical cells. Because the Ekman transport is driven by the trade winds, the ocean circulation is coupled to the Hadley circulation and the ITCZ. The heat transport by the subtropical cells decreases the amount of heat that needs to be transported by the Hadley circulation because both are transporting heat in the same direction. Green and Schneider have both quantified (in their model frameworks) how much the ocean circulation damps the atmospheric circulation. Here, comparing the models with the SOM and the RGO (which includes a representation of Ekman

transport) is an additional test of Green's/Schneider's arguments. Green/Schneider's work suggests that the model with the RGO should have less of an ITCZ shift in comparison to the model with the SOM, and that is exactly the result presented here. This manuscript provides a nice confirmation of these previous results, and should be presented as such. If possible, it would be useful to calculate how much the ocean damps the atmospheric heat transport in the RGO as compared to the SOM (a factor of 4 in Green and 3 in Schneider). I also suggest adding discussion of Green's/Schneider's results to the introduction and summary/conclusions, and possibly to a couple of these other relevant sections in the text: L168-169, L171-172, L181-183, L270-274.

Minor Comment 1 (Figure quality): Please make the figures more easily digestible for the reader by placing key information about what is displayed in each panel both on the panel itself and in the caption. The closer the visual information is to a description of what it is, the less hunting the reader needs to do and the less likely the reader will become confused or give up. This includes: labels with units on the colorbars themselves (all colorbars lack this) and labels on the panels that state which experiment and model are shown in that panel (e.g., for Fig 1a. could have a label that says "NOAA SST, Pacific" or Fig 4a. could have a label that says "Forced_slab"). Also, in figures with maps it appears that much of Central America is missing. Is this missing in the model or just an artifact of the plotting?

Minor Comment 2 (L23-26): A bit deeper explanation of the mechanism behind this heuristic would be useful for the readers who are not familiar with this energetic constraint on tropical precipitation. This will probably be useful when adding the explanations in Green/Schneider. There are also numerous additional citations that could (should?) be added here. A couple are: Zhang and Delworth (2005) (DOI: 10.1175/JCLI3460.1), Broccoli et al. (2006) (DOI: 10.1029/2005GL024546), Schneider et al. (2014) (DOI: 10.1038/nature13636), Bischoff and Schneider (2014) (DOI: 10.1175/JCLI-D-13-00650.1), Seo et al. (2014) (DOI: 10.1175/JCLI-D-13-00691.1), Woelfle et al. (2015) (DOI: 10.1002/2015GL063372). Additional relevant references

can be found in these papers.

Minor Comment 3 (L59-60): This sentence is slightly misleading – there has been much work done using a hierarchy of atmospheric models to test these ideas. A couple are: Shaw et al. (2015) (DOI: 10.1002/2015GL0660270), Seo et al. (2014) (DOI: 10.1175/JCLI-D-13-00691.1), Maroon et al. (2014) (DOI: 10.1175/JCLI-D-14-00188.1). Please rephrase to state more clearly that fewer studies have examined this topic in a hierarchy of ocean models.

Minor Comment 4 (L60-65): There are an additional two studies that have tested similar ideas as Kay and Hawcroft: Mechoso et al. (2016) (DOI: 10.1002/2016GL071150) and Tomas et al. (2016) (DOI: 10.1175/JCLI-D-15-0651.1)

Minor Comment 5 (L79): Please state in a bit more depth exactly what the simplified atmospheric physics are. As the complexity of atmospheric physics affect the magnitude of an ITCZ shift to extratropical forcing, an additional sentence briefly stating what these simplifications are is warranted.

Minor Comment 6 (L84,87): What are the ocean heat flux corrections derived from? Observational datasets? A fully-coupled version of this model?

Minor Comment 7 (L210-211, Figure 9/10): Because of the chosen figure scale, the eastward/westward anomalies referenced in this sentence are not visible, which makes this statement confusing.

Technical Corrections

L241, 243, 246: Incorrect figure references L249, 298: extra commas in citations L270-271: awkward grammar L291: be careful using the word significant if not conducting statistical significance tests Figure 11: The caption incorrectly identifies 2 panels when there is only 1.

---

## Referee Comment (RC2) · Anonymous Referee #2 · 1 Jan 2018

General Comments:

In this paper, the authors employ a climate model hierarchy to understand the climate response to an idealized interhemispheric thermal gradient (ITG). The model hierarchy consists of an atmospheric general circulation model (AGCM) under two coupled configurations. In the first configuration, the AGCM is coupled to a slab ocean model everywhere on the globe. In the second configuration, the AGCM is coupled to a slab ocean model everywhere except in the tropics where it is coupled to a reduced gravity ocean model to yield a total of four simulations. The two configurations are run under two scenarios: a scenario in which no forcing perturbation is applied and in a scenario

in which the idealized ITG is imposed to yield a set of four simulations. Using the four simulations, the role of ocean dynamics in the climate response to extratropical forcing perturbations is studied. The authors demonstrate that including tropical ocean dynamics mutes ITCZ shifts in response to imposed ITGs. Further, the authors show that the tropical seasonal cycle is intensified and in response the ENSO is weakened when ocean dynamics are included.

The paper nicely complements the recent results of Kay et al (2016) who showed a similar climate response to an imposed extratropical forcing in a climate model hierarchy and emphasizes the importance of ocean dynamics in determining the ITCZ response to forcing perturbations. The paper is structurally and logically well organized. I suggest publication with the following revisions.

Specific Comments:

Major Comment

My major comment on this paper pertains to the lack of discussion on the recent relevant results of Green and Marshall (2017) and also, as pointed by Reviewer 1, of Schneider (2017) that provide a physical pathway for how ocean coupling mutes the ITCZ response to interhemispheric energy perturbations. The findings of these papers and their relevance to this study are described adequately in Major Comment 1 by Reviewer 1, so I will skip repeating the discussion here. I however have a few minor and a number of technical comments that I list below.

Minor Comments

Line 79: Consider including a sentence or two describing the simplified physics

Lines 51, 128, 150, 216, 291: Consider using the word 'significant' only when referring to statistical significance. Otherwise, I suggest replacing with synonyms like 'considerable' etc.,

Technical comments

Line 33: trough -> through

Line 46: being -> with

Line 63: being -> with

Line 74: find -> found

Line 90: validate its results comparing -> validate its results by comparing

Line 106: That means that, for momentum and heat fluxes, the oceanic and atmospheric components of the model exchange anomalies computed relative to their own model annual mean → In this strategy, the oceanic and atmospheric components of the model exchange momentum and heat flux anomalies computed relative to their own model annual mean

Line 108: superimposed to -> superimposed on

Line 108: wide -> width

Line 113: analogous -> analogues

Line 116: in the Control the simulated annual mean SST -> the annual mean SST in the control simulation

Line 125: than the observed and with the -> than the observed, with the

Line 126: as do in the observations -> as it does in the observations

Line 132: pattern consists in cooling -> patterns consists of cooling

Line 136: .asymmetric -> asymmetric

Line 136: is superposed to a -> is superposed on a

Line 155: focus in -> focus on

Line 155: produced to → produced in

Line 201: being September-November (SON) the period of strongest cooling and the June-August (JJA) period the one -> September-November (SON) being the period of strongest cooling and June-August (JJA) being the period

Line 202: this negative -> the negative

Line 270: being the signal produced with the RGO coupling weaker in terms of annual means -> with the signal produced in the RGO coupling case being weaker in terms of annual means

Line 215: Figure 10a shows SST anomalies and not wind anomalies. Please refer to appropriate figure.

Line 255: Timmermann et al., (2007) is missing from the list of references.

All figures: Please include headings for figure panels as visual aids

Figure 1 and 2: increase font size for x and y labels in figures 1 and 2.

Figure 11: The figure panel corresponding to Forced_slab (Figure 11a) is missing

---

## Author Comment (AC1) · 1 Feb 2018

We thank the reviewer for his/her constructive comments and suggestions.

We have given full consideration to the comments in the revised manuscript which includes: a discussion of the results of Green and Marshall (2017) and Schneider (2017) as well as figure modifications to make them easily interpreted by the reader.

Please find below a point-by-point reply to the questions raised. A marked-up manuscript version (with tracked changes) converted into a pdf is also uploaded as a supplement.

Anonymous Referee #1

General Comments

The main motivation of this manuscript titled "Sensitivity of the tropical climate to an interhemispheric thermal gradient: the role of tropical ocean dynamics" is to understand how the addition of a simple dynamic ocean model in the tropics affects the climate response to idealized extratropical forcing with different signs in each hemisphere. Four coupled simulations are presented, two that are coupled to a slab ocean model (SOM) and two that are coupled to a reduced gravity ocean (RGO) model. The extratropical forcing is applied to one simulation in each of these model pairs. The main results are that the tropical precipitation shift in response to the forcing is weaker in the model with the RGO and that the seasonal cycle in the tropics has large change in response to the forcing in the model with the RGO but not the SOM. To my best knowledge, the examination of ITCZ shifts to extratropical thermal forcing with a climate model coupled to a reduced gravity ocean has not been done yet. This examination is timely given the results presented in Kay et al. (2016) and Hawcroft et al. (2017) that show a clear role for ocean dynamics in affecting tropical precipitation shifts. That said, this manuscript misses a few key studies from the past year that are very relevant: Green and Marshall (2017) and Schneider (2017). These two studies provide a mechanistic explanation for how tropical ocean circulation damps an ITCZ shift through coupling of the ocean and atmosphere by wind stress (see comments below). The results in this manuscript support the arguments in these two papers, and I believe that further discussion in light of Green's and Schneider's arguments would strengthen this manuscript. Otherwise, I find this manuscript clearly argued and written, and I only have a few other minor comments, mostly regarding figure quality and additional related literature. I recommend this paper for acceptance pending a major revision to include discussion of Green and Schneider and the minor issues listed below.

Specific Comments

Major comment 1:

Green and Marshall (2017) (DOI: 10.1175/JCLI-D-16-0818.1) and Schneider (2017) (DOI: 10.1002/2017GL075817) have both shown that an ITCZ shift is damped by tropical ocean circulation, specifically by meridional heat transport through Ekman transport by the subtropical cells. Because the Ekman transport is driven by the trade winds, the ocean circulation is coupled to the Hadley circulation and the ITCZ. The heat transport by the subtropical cells decreases the amount of heat that needs to be transported by the Hadley circulation because both are transporting heat in the same direction. Green and Schneider have both quantified (in their model frameworks) how much the ocean circulation damps the atmospheric circulation. Here, comparing the models with the SOM and the RGO (which includes a representation of Ekman transport) is an additional test of Green's/Schneider's arguments. Green/Schneider's work suggests that the model with the RGO should have less of an ITCZ shift in comparison to the model with the SOM, and that is exactly the result presented here. This manuscript provides a nice confirmation of these previous results, and should be presented as such. If possible, it would be useful to calculate how much the ocean damps the atmospheric heat transport in the RGO as compared to the SOM (a factor of 4 in Green and 3 in Schneider). I also suggest adding discussion of Green's/Schneider's results to the introduction and summary/conclusions, and possibly to a couple of these other relevant sections in the text: L168-169, L171-172, L181-183, L270-274.

***************************************************************************** Agreed. Discussions about the results of the two mentioned papers have been included in the Introduction and Summary and Conclusions sections. Also, the damping of the atmospheric heat transport and energy flux equator shift have been calculated (1.9 and 1.5, respectively) and included in the text, in section "Results, annual means". *****************************************************************************

Minor Comment 1 (Figure quality): Please make the figures more easily digestible for the reader by placing key information about what is displayed in each panel both on

the panel itself and in the caption. The closer the visual information is to a description of what it is, the less hunting the reader needs to do and the less likely the reader will become confused or give up. This includes: labels with units on the colorbars themselves (all colorbars lack this) and labels on the panels that state which experiment and model are shown in that panel (e.g., for Fig 1a. could have a label that says "NOAA SST, Pacific" or Fig 4a. could have a label that says "Forced_slab"). Also, in figures with maps it appears that much of Central America is missing. Is this missing in the model or just an artifact of the plotting?

******************************************************************************* Agreed. All the Figures have been modified to include: headings with experiment name or other key relevant information and the corresponding units in the colorbars. Regarding the longitude-latitude maps: The land-sea mask plotted is the one that the model uses (Central America is missing with this model resolution). We added 1 sentence in the text clarifying this. *******************************************************************************

Minor Comment 2 (L23-26): A bit deeper explanation of the mechanism behind this heuristic would be useful for the readers who are not familiar with this energetic constraint on tropical precipitation. This will probably be useful when adding the explanations in Green/Schneider. There are also numerous additional citations that could (should?) be added here. A couple are: Zhang and Delworth (2005) (DOI: 10.1175/JCLI3460.1), Broccoli et al. (2006) (DOI: 10.1029/2005GL024546), Schneider et al. (2014) (DOI: 10.1038/nature13636), Bischoff and Schneider (2014) (DOI: 10.1175/JCLI-D-13-00650.1), Seo et al. (2014) (DOI: 10.1175/JCLI-D-13-00691.1), Woelfle et al. (2015) (DOI: 10.1002/2015GL063372). Additional relevant references can be found in these papers.

******************************************************************************* Agreed. A couple of sentences have been added to clarify this in the Introduction section. Also, the references of Broccoli et al. (2016) and Schneider et al. (2014) have been incorporated to the manuscript. *******************************************************************************

Minor Comment 3 (L59-60): This sentence is slightly misleading – there has been much work done using a hierarchy of atmospheric models to test these ideas. A couple are: Shaw et al. (2015) (DOI: 10.1002/2015GL0660270), Seo et al. (2014) (DOI: 10.1175/JCLI-D-13-00691.1), Maroon et al. (2014) (DOI: 10.1175/JCLI-D-14-00188.1). Please rephrase to state more clearly that fewer studies have examined this topic in a hierarchy of ocean models.

*************************************************************************** Agreed, the sentence has been rephrased. ***********************************************************************************

Minor Comment 4 (L60-65): There are an additional two studies that have tested similar ideas as Kay and Hawcroft: Mechoso et al. (2016) (DOI: 10.1002/2016GL071150) and Tomas et al. (2016) (DOI: 10.1175/JCLI-D-15-0651.1)
* * *
Agreed, Tomas et al. (2016) has been added as reference. ***************************************************************************

Minor Comment 5 (L79): Please state in a bit more depth exactly what the simplified atmospheric physics are. As the complexity of atmospheric physics affect the magnitude of an ITCZ shift to extratropical forcing, an additional sentence briefly stating what these simplifications are is warranted.

*************************************************************************** Agreed. The following text has been added to the model description: "The model includes parameterizations of: large-scale condensation, shallow and deep convection, shortwave radiation (using 2 spectral bands), longwave radiation (using 4 spectral bands), surface fluxes of momentum, heat and moisture, and vertical diffusion" ***************************************************************************

Minor Comment 6 (L84,87): What are the ocean heat flux corrections derived from? Observational datasets? A fully-coupled version of this model?

\*\*\*\*\*\*\*\*\*\*\*\*\*\*\*\*\*\*\*\*\*\*\*\*\*\*\*\*\*\*\*\*\*\*\*\*\*\*\*\*\*\*\*\*\*\*\*\*\*\*\*\*\*\*\*\*\*\*\*\*\*\*\*\*\*\*\*\*\*\*\*\*\*\*\*\* The correction is derived from a previous model simulation in which all the parameters and settings are identical but in which observed SSTs are prescribed (in this case observed SSTs for the 30-years period 1979-2008). In the text the sentence has been modified to include this informations in the following way: "We analyze the outcomes of coupling the AGCM with two ocean models of different complexity. In the first configuration the AGCM is coupled with a slab ocean model; a monthly-varying ocean heat flux correction (derived from a previous 30-year model integration with identical settings but with prescribed observed SSTs) is imposed in order to keep the simulated SST close to present-day conditions." \*\*\*\*\*\*\*\*\*\*\*\*\*\*\*\*\*\*\*\*\*\*\*\*\*\*\*\*\*\*\*\*\*\*\*\*\*\*\*\*\*\*\*\*\*\*\*\*\*\*\*\*\*\*\*\*\*\*\*\*\*\*\*\*\*\*\*\*\*\*\*\*\*\*\*\*\*\*\*\*

Minor Comment 7 (L210-211, Figure 9/10): Because of the chosen figure scale, the eastward/westward anomalies referenced in this sentence are not visible, which makes this statement confusing.

\*\*\*\*\*\*\*\*\*\*\*\*\*\*\*\*\*\*\*\*\*\*\*\*\*\*\*\*\*\*\*\*\*\*\*\*\*\*\*\*\*\*\*\*\*\*\*\*\*\*\*\*\*\*\*\*\*\*\*\*\*\*\*\*\*\*\*\*\*\*\*\*\*\*\*\* Disagreed. The anomalies referred in the text are easily seen in the mentioned Figures. \*\*\*\*\*\*\*\*\*\*\*\*\*\*\*\*\*\*\*\*\*\*\*\*\*\*\*\*\*\*\*\*\*\*\*\*\*\*\*\*\*\*\*\*\*\*\*\*\*\*\*\*\*\*\*\*\*\*\*\*\*\*\*\*\*\*\*\*\*\*\*\*\*\*

Technical Corrections

L241, 243, 246: Incorrect figure references

\*\*\*\*\*\*\*\*\*\*\*\*\*\*\*\*\*\*\*\*\*\*\*\*\*\*\*\*\*\*\*\*\*\*\*\*\*\*\*\*\*\*\*\*\*\*\*\*\*\*\*\*\*\*\*\*\*\*\*\*\*\*\*\*\*\*\*\*\*\*\*\*\*\*\*\*\*\*\*\*     Agreed
\*\*\*\*\*\*\*\*\*\*\*\*\*\*\*\*\*\*\*\*\*\*\*\*\*\*\*\*\*\*\*\*\*\*\*\*\*\*\*\*\*\*\*\*\*\*\*\*\*\*\*\*\*\*\*\*\*\*\*\*\*\*\*\*\*\*\*\*\*\*\*\*\*\*\*\*\*\*\*\*

L249, 298: extra commas in citations

\*\*\*\*\*\*\*\*\*\*\*\*\*\*\*\*\*\*\*\*\*\*\*\*\*\*\*\*\*\*\*\*\*\*\*\*\*\*\*\*\*\*\*\*\*\*\*\*\*\*\*\*\*\*\*\*\*\*\*\*\*\*\*\*\*\*\*\*\*\*\*\*\*\*\*\*\*\*\*\*     Agreed
\*\*\*\*\*\*\*\*\*\*\*\*\*\*\*\*\*\*\*\*\*\*\*\*\*\*\*\*\*\*\*\*\*\*\*\*\*\*\*\*\*\*\*\*\*\*\*\*\*\*\*\*\*\*\*\*\*\*\*\*\*\*\*\*\*\*\*\*\*\*\*\*\*\*\*\*\*\*\*\*

L270- 271: awkward grammar

\*\*\*\*\*\*\*\*\*\*\*\*\*\*\*\*\*\*\*\*\*\*\*\*\*\*\*\*\*\*\*\*\*\*\*\*\*\*\*\*\*\*\*\*\*\*\*\*\*\*\*\*\*\*\*\*\*\*\*\*\*\*\*\*\*\*\*\*\*\*\*\*\*\*\*\*\*\*\*\*     Agreed

[Figure]

\*\*\*\*\*\*\*\*\*\*\*\*\*\*\*\*\*\*\*\*\*\*\*\*\*\*\*\*\*\*\*\*\*\*\*\*\*\*\*\*\*\*\*\*\*\*\*\*\*\*\*\*\*\*\*\*\*\*\*\*\*\*\*\*\*\*\*\*\*\*\*\*\*\*\*\*

L291: be careful using the word significant if not conducting statistical significance tests

\*\*\*\*\*\*\*\*\*\*\*\*\*\*\*\*\*\*\*\*\*\*\*\*\*\*\*\*\*\*\*\*\*\*\*\*\*\*\*\*\*\*\*\*\*\*\*\*\*\*\*\*\*\*\*\*\*\*\*\*\*\*\*\*\*\*\*\*\*\*\*\*\*\*\*\*                                        Agreed

\*\*\*\*\*\*\*\*\*\*\*\*\*\*\*\*\*\*\*\*\*\*\*\*\*\*\*\*\*\*\*\*\*\*\*\*\*\*\*\*\*\*\*\*\*\*\*\*\*\*\*\*\*\*\*\*\*\*\*\*\*\*\*\*\*\*\*\*\*\*\*\*\*\*\*\*

Figure 11: The caption incorrectly identifies 2 panels when there is only 1.

\*\*\*\*\*\*\*\*\*\*\*\*\*\*\*\*\*\*\*\*\*\*\*\*\*\*\*\*\*\*\*\*\*\*\*\*\*\*\*\*\*\*\*\*\*\*\*\*\*\*\*\*\*\*\*\*\*\*\*\*\*\*\*\*\*\*\*\*\*\*\*\*\*\*\*\* Agreed, the caption
has been modified. \*\*\*\*\*\*\*\*\*\*\*\*\*\*\*\*\*\*\*\*\*\*\*\*\*\*\*\*\*\*\*\*\*\*\*\*\*\*\*\*\*\*\*\*\*\*\*\*\*\*\*\*\*\*\*\*\*\*\*\*\*\*\*\*\*\*\*\*\*\*\*\*

Please also note the supplement to this comment:
https://www.earth-syst-dynam-discuss.net/esd-2017-113/esd-2017-113-AC1-
supplement.pdf

**Supplement:**

[revised manuscript text omitted]

---

## Author Comment (AC2) · 1 Feb 2018

We thank the reviewer for his/her constructive comments and suggestions.

We have given full consideration to the comments in the revised manuscript which includes: a discussion of the results of Green and Marshall (2017) and Schneider (2017) as well as figure modifications to make them easily interpreted by the reader.

Please find below a point-by-point reply to the questions raised. A marked-up manuscript version (with tracked changes) converted into a pdf is also uploaded as a supplement.

[Figure]

Anonymous Referee #2

General Comments: In this paper, the authors employ a climate model hierarchy to understand the climate response to an idealized interhemispheric thermal gradient (ITG). The model hierarchy consists of an atmospheric general circulation model (AGCM) under two coupled configurations. In the first configuration, the AGCM is coupled to a slab ocean model everywhere on the globe. In the second configuration, the AGCM is coupled to a slab ocean model everywhere except in the tropics where it is coupled to a reduced gravity ocean model to yield a total of four simulations. The two configurations are run under two scenarios: a scenario in which no forcing perturbation is applied and in a scenario in which the idealized ITG is imposed to yield a set of four simulations. Using the four simulations, the role of ocean dynamics in the climate response to extratropical forcing perturbations is studied. The authors demonstrate that including tropical ocean dynamics mutes ITCZ shifts in response to imposed ITGs. Further, the authors show that the tropical seasonal cycle is intensified and in response the ENSO is weakened when ocean dynamics are included.

The paper nicely complements the recent results of Kay et al (2016) who showed a similar climate response to an imposed extratropical forcing in a climate model hierarchy and emphasizes the importance of ocean dynamics in determining the ITCZ response to forcing perturbations. The paper is structurally and logically well organized. I suggest publication with the following revisions.

Specific Comments:

Major Comment

My major comment on this paper pertains to the lack of discussion on the recent relevant results of Green and Marshall (2017) and also, as pointed by Reviewer 1, of Schneider (2017) that provide a physical pathway for how ocean coupling mutes the ITCZ response to interhemispheric energy perturbations. The findings of these papers and their relevance to this study are described adequately in Major Comment 1 by Re-

viewer 1, so I will skip repeating the discussion here. I however have a few minor and a number of technical comments that I list below.

\*\*\*\*\*\*\*\*\*\*\*\*\*\*\*\*\*\*\*\*\*\*\*\*\*\*\*\*\*\*\*\*\*\*\*\*\*\*\*\*\*\*\*\*\*\*\*\*\*\*\*\*\*\*\*\*\*\*\*\*\*\*\*\*\*\*\*\*\*\*\*\*\*\*\* Agreed.
The results of the two mentioned papers have been included in the Introduction and Summary and Conclusions section.
\*\*\*\*\*\*\*\*\*\*\*\*\*\*\*\*\*\*\*\*\*\*\*\*\*\*\*\*\*\*\*\*\*\*\*\*\*\*\*\*\*\*\*\*\*\*\*\*\*\*\*\*\*\*\*\*\*\*\*\*\*\*\*\*\*\*\*\*\*\*\*\*\*\*\*

Minor Comments

Line 79: Consider including a sentence or two describing the simplified physics

\*\*\*\*\*\*\*\*\*\*\*\*\*\*\*\*\*\*\*\*\*\*\*\*\*\*\*\*\*\*\*\*\*\*\*\*\*\*\*\*\*\*\*\*\*\*\*\*\*\*\*\*\*\*\*\*\*\*\*\*\*\*\*\*\*\*\*\*\*\*\*\* Agreed. The following text has been added to the model description: "The model includes parameterizations of: large-scale condensation, shallow and deep convection, shortwave radiation (using 2 spectral bands), longwave radiation (using 4 spectral bands), surface fluxes of momentum, heat and moisture, and vertical diffusion"
\*\*\*\*\*\*\*\*\*\*\*\*\*\*\*\*\*\*\*\*\*\*\*\*\*\*\*\*\*\*\*\*\*\*\*\*\*\*\*\*\*\*\*\*\*\*\*\*\*\*\*\*\*\*\*\*\*\*\*\*\*\*\*\*\*\*\*\*\*\*\*\*\*\*\*

Lines 51, 128, 150, 216, 291: Consider using the word 'significant' only when referring to statistical significance. Otherwise, I suggest replacing with synonyms like 'considerable' etc., Technical comments

\*\*\*\*\*\*\*\*\*\*\*\*\*\*\*\*\*\*\*\*\*\*\*\*\*\*\*\*\*\*\*\*\*\*\*\*\*\*\*\*\*\*\*\*\*\*\*\*\*\*\*\*\*\*\*\*\*\*\*\*\*\*\*\*\*\*\*\*\*\*\*\*\*\*\* Agreed
\*\*\*\*\*\*\*\*\*\*\*\*\*\*\*\*\*\*\*\*\*\*\*\*\*\*\*\*\*\*\*\*\*\*\*\*\*\*\*\*\*\*\*\*\*\*\*\*\*\*\*\*\*\*\*\*\*\*\*\*\*\*\*\*\*\*\*\*\*\*\*\*\*\*\*

Line 33: trough -> through

\*\*\*\*\*\*\*\*\*\*\*\*\*\*\*\*\*\*\*\*\*\*\*\*\*\*\*\*\*\*\*\*\*\*\*\*\*\*\*\*\*\*\*\*\*\*\*\*\*\*\*\*\*\*\*\*\*\*\*\*\*\*\*\*\*\*\*\*\*\*\*\*\*\*\* Agreed
\*\*\*\*\*\*\*\*\*\*\*\*\*\*\*\*\*\*\*\*\*\*\*\*\*\*\*\*\*\*\*\*\*\*\*\*\*\*\*\*\*\*\*\*\*\*\*\*\*\*\*\*\*\*\*\*\*\*\*\*\*\*\*\*\*\*\*\*\*\*\*\*\*\*\*

Line 46: being -> with

\*\*\*\*\*\*\*\*\*\*\*\*\*\*\*\*\*\*\*\*\*\*\*\*\*\*\*\*\*\*\*\*\*\*\*\*\*\*\*\*\*\*\*\*\*\*\*\*\*\*\*\*\*\*\*\*\*\*\*\*\*\*\*\*\*\*\*\*\*\*\*\*\*\*\* Agreed
\*\*\*\*\*\*\*\*\*\*\*\*\*\*\*\*\*\*\*\*\*\*\*\*\*\*\*\*\*\*\*\*\*\*\*\*\*\*\*\*\*\*\*\*\*\*\*\*\*\*\*\*\*\*\*\*\*\*\*\*\*\*\*\*\*\*\*\*\*\*\*\*\*\*\*

Line 63: being -> with

\*\*\*\*\*\*\*\*\*\*\*\*\*\*\*\*\*\*\*\*\*\*\*\*\*\*\*\*\*\*\*\*\*\*\*\*\*\*\*\*\*\*\*\*\*\*\*\*\*\*\*\*\*\*\*\*\*\*\*\*\*\*\*\*\*\*\*\*\*\*\*\*\*\*\*\*\*\*\*\*\*\*   Agreed

\*\*\*\*\*\*\*\*\*\*\*\*\*\*\*\*\*\*\*\*\*\*\*\*\*\*\*\*\*\*\*\*\*\*\*\*\*\*\*\*\*\*\*\*\*\*\*\*\*\*\*\*\*\*\*\*\*\*\*\*\*\*\*\*\*\*\*\*\*\*\*\*\*\*\*\*\*\*\*\*\*\*

Line 74: find -> found

\*\*\*\*\*\*\*\*\*\*\*\*\*\*\*\*\*\*\*\*\*\*\*\*\*\*\*\*\*\*\*\*\*\*\*\*\*\*\*\*\*\*\*\*\*\*\*\*\*\*\*\*\*\*\*\*\*\*\*\*\*\*\*\*\*\*\*\*\*\*\*\*\*\*\*\*\*\*\*\*\*\*   Agreed

\*\*\*\*\*\*\*\*\*\*\*\*\*\*\*\*\*\*\*\*\*\*\*\*\*\*\*\*\*\*\*\*\*\*\*\*\*\*\*\*\*\*\*\*\*\*\*\*\*\*\*\*\*\*\*\*\*\*\*\*\*\*\*\*\*\*\*\*\*\*\*\*\*\*\*\*\*\*\*\*\*\*

Line 90: validate its results comparing -> validate its results by comparing

\*\*\*\*\*\*\*\*\*\*\*\*\*\*\*\*\*\*\*\*\*\*\*\*\*\*\*\*\*\*\*\*\*\*\*\*\*\*\*\*\*\*\*\*\*\*\*\*\*\*\*\*\*\*\*\*\*\*\*\*\*\*\*\*\*\*\*\*\*\*\*\*\*\*\*\*\*\*\*\*\*\*   Agreed

\*\*\*\*\*\*\*\*\*\*\*\*\*\*\*\*\*\*\*\*\*\*\*\*\*\*\*\*\*\*\*\*\*\*\*\*\*\*\*\*\*\*\*\*\*\*\*\*\*\*\*\*\*\*\*\*\*\*\*\*\*\*\*\*\*\*\*\*\*\*\*\*\*\*\*\*\*\*\*\*\*\*

Line 106: That means that, for momentum and heat fluxes, the oceanic and atmospheric components of the model exchange anomalies computed relative to their own model annual mean → In this strategy, the oceanic and atmospheric components of the model exchange momentum and heat flux anomalies computed relative to their own model annual mean

\*\*\*\*\*\*\*\*\*\*\*\*\*\*\*\*\*\*\*\*\*\*\*\*\*\*\*\*\*\*\*\*\*\*\*\*\*\*\*\*\*\*\*\*\*\*\*\*\*\*\*\*\*\*\*\*\*\*\*\*\*\*\*\*\*\*\*\*\*\*\*\*\*\*\*\*\*\*\*\*\*\*   Agreed

\*\*\*\*\*\*\*\*\*\*\*\*\*\*\*\*\*\*\*\*\*\*\*\*\*\*\*\*\*\*\*\*\*\*\*\*\*\*\*\*\*\*\*\*\*\*\*\*\*\*\*\*\*\*\*\*\*\*\*\*\*\*\*\*\*\*\*\*\*\*\*\*\*\*\*\*\*\*\*\*\*\*

Line 108: superimposed to -> superimposed on

\*\*\*\*\*\*\*\*\*\*\*\*\*\*\*\*\*\*\*\*\*\*\*\*\*\*\*\*\*\*\*\*\*\*\*\*\*\*\*\*\*\*\*\*\*\*\*\*\*\*\*\*\*\*\*\*\*\*\*\*\*\*\*\*\*\*\*\*\*\*\*\*\*\*\*\*\*\*\*\*\*\*   Agreed

\*\*\*\*\*\*\*\*\*\*\*\*\*\*\*\*\*\*\*\*\*\*\*\*\*\*\*\*\*\*\*\*\*\*\*\*\*\*\*\*\*\*\*\*\*\*\*\*\*\*\*\*\*\*\*\*\*\*\*\*\*\*\*\*\*\*\*\*\*\*\*\*\*\*\*\*\*\*\*\*\*\*

Line 108: wide -> width

\*\*\*\*\*\*\*\*\*\*\*\*\*\*\*\*\*\*\*\*\*\*\*\*\*\*\*\*\*\*\*\*\*\*\*\*\*\*\*\*\*\*\*\*\*\*\*\*\*\*\*\*\*\*\*\*\*\*\*\*\*\*\*\*\*\*\*\*\*\*\*\*\*\*\*\*\*\*\*\*\*\*   Agreed

\*\*\*\*\*\*\*\*\*\*\*\*\*\*\*\*\*\*\*\*\*\*\*\*\*\*\*\*\*\*\*\*\*\*\*\*\*\*\*\*\*\*\*\*\*\*\*\*\*\*\*\*\*\*\*\*\*\*\*\*\*\*\*\*\*\*\*\*\*\*\*\*\*\*\*\*\*\*\*\*\*\*

Line 113: analogous -> analogues

\*\*\*\*\*\*\*\*\*\*\*\*\*\*\*\*\*\*\*\*\*\*\*\*\*\*\*\*\*\*\*\*\*\*\*\*\*\*\*\*\*\*\*\*\*\*\*\*\*\*\*\*\*\*\*\*\*\*\*\*\*\*\*\*\*\*\*\*\*\*\*\*\*\*\*\*\*\*\*\*\*\*   Agreed
* * *
Line 116: in the Control the simulated annual mean SST -> the annual mean SST in
the control simulation

****************************************************************************** Agreed
* * *
Line 125: than the observed and with the -> than the observed, with the

****************************************************************************** Agreed
* * *
Line 126: as do in the observations -> as it does in the observations

****************************************************************************** Agreed
* * *
Line 132: pattern consists in cooling -> patterns consists of cooling

****************************************************************************** Agreed
* * *
Line 136: .asymmetric -> asymmetric

****************************************************************************** Agreed
* * *
Line 136: is superposed to a -> is superposed on a

****************************************************************************** Agreed
* * *
Line 155: focus in -> focus on

****************************************************************************** Agreed
* * *
Line 155: produced to → produced in Line 201: being September-November (SON) the period of strongest cooling and the June-August (JJA) period the one -> September-November (SON) being the period of strongest cooling and June-August (JJA) being the period

\*\*\*\*\*\*\*\*\*\*\*\*\*\*\*\*\*\*\*\*\*\*\*\*\*\*\*\*\*\*\*\*\*\*\*\*\*\*\*\*\*\*\*\*\*\*\*\*\*\*\*\*\*\*\*\*\*\*\*\*\*\*\*\*\*\*\*\*\*\*\*\*\*\*\*\*\*\*\*\*\*\*\*\*\*\* Agreed
\*\*\*\*\*\*\*\*\*\*\*\*\*\*\*\*\*\*\*\*\*\*\*\*\*\*\*\*\*\*\*\*\*\*\*\*\*\*\*\*\*\*\*\*\*\*\*\*\*\*\*\*\*\*\*\*\*\*\*\*\*\*\*\*\*\*\*\*\*\*\*\*\*\*\*\*\*\*\*\*\*\*\*\*\*\*

Line 202: this negative -> the negative

\*\*\*\*\*\*\*\*\*\*\*\*\*\*\*\*\*\*\*\*\*\*\*\*\*\*\*\*\*\*\*\*\*\*\*\*\*\*\*\*\*\*\*\*\*\*\*\*\*\*\*\*\*\*\*\*\*\*\*\*\*\*\*\*\*\*\*\*\*\*\*\*\*\*\*\*\*\*\*\*\*\*\*\*\*\* Agreed
\*\*\*\*\*\*\*\*\*\*\*\*\*\*\*\*\*\*\*\*\*\*\*\*\*\*\*\*\*\*\*\*\*\*\*\*\*\*\*\*\*\*\*\*\*\*\*\*\*\*\*\*\*\*\*\*\*\*\*\*\*\*\*\*\*\*\*\*\*\*\*\*\*\*\*\*\*\*\*\*\*\*\*\*\*\*

Line 270: being the signal produced with the RGO coupling weaker in terms of annual means -> with the signal produced in the RGO coupling case being weaker in terms of annual means

\*\*\*\*\*\*\*\*\*\*\*\*\*\*\*\*\*\*\*\*\*\*\*\*\*\*\*\*\*\*\*\*\*\*\*\*\*\*\*\*\*\*\*\*\*\*\*\*\*\*\*\*\*\*\*\*\*\*\*\*\*\*\*\*\*\*\*\*\*\*\*\*\*\*\*\*\*\*\*\*\*\*\*\*\*\* Agreed
\*\*\*\*\*\*\*\*\*\*\*\*\*\*\*\*\*\*\*\*\*\*\*\*\*\*\*\*\*\*\*\*\*\*\*\*\*\*\*\*\*\*\*\*\*\*\*\*\*\*\*\*\*\*\*\*\*\*\*\*\*\*\*\*\*\*\*\*\*\*\*\*\*\*\*\*\*\*\*\*\*\*\*\*\*\*

Line 215: Figure 10a shows SST anomalies and not wind anomalies. Please refer to appropriate figure. Disagreed, the Figure reference is correct.

Line 255: Timmermann et al., (2007) is missing from the list of references.

\*\*\*\*\*\*\*\*\*\*\*\*\*\*\*\*\*\*\*\*\*\*\*\*\*\*\*\*\*\*\*\*\*\*\*\*\*\*\*\*\*\*\*\*\*\*\*\*\*\*\*\*\*\*\*\*\*\*\*\*\*\*\*\*\*\*\*\*\*\*\*\*\*\*\*\*\*\*\*\*\* Agreed, the reference has been added. \*\*\*\*\*\*\*\*\*\*\*\*\*\*\*\*\*\*\*\*\*\*\*\*\*\*\*\*\*\*\*\*\*\*\*\*\*\*\*\*\*\*\*\*\*\*\*\*\*\*\*\*\*\*\*\*\*\*\*\*\*\*\*\*\*\*\*\*\*\*\*\*

All figures: Please include headings for figure panels as visual aids

\*\*\*\*\*\*\*\*\*\*\*\*\*\*\*\*\*\*\*\*\*\*\*\*\*\*\*\*\*\*\*\*\*\*\*\*\*\*\*\*\*\*\*\*\*\*\*\*\*\*\*\*\*\*\*\*\*\*\*\*\*\*\*\*\*\*\*\*\*\*\*\*\*\*\*\*\*\*\*\*\* Agreed, headings indicating the experiment or key information has been added. \*\*\*\*\*\*\*\*\*\*\*\*\*\*\*\*\*\*\*\*\*\*\*\*\*\*\*\*\*\*\*\*\*\*\*\*\*\*\*\*\*\*\*\*\*\*\*\*\*\*\*\*\*\*\*\*\*\*\*\*\*\*\*\*\*\*\*\*\*\*\*\*\*\*\*\*\*

Figure 1 and 2: increase font size for x and y labels in figures 1 and 2.

\*\*\*\*\*\*\*\*\*\*\*\*\*\*\*\*\*\*\*\*\*\*\*\*\*\*\*\*\*\*\*\*\*\*\*\*\*\*\*\*\*\*\*\*\*\*\*\*\*\*\*\*\*\*\*\*\*\*\*\*\*\*\*\*\*\*\*\*\*\*\*\*\*\*\*\*\*\*\*\*\*\*\* Agreed
\*\*\*\*\*\*\*\*\*\*\*\*\*\*\*\*\*\*\*\*\*\*\*\*\*\*\*\*\*\*\*\*\*\*\*\*\*\*\*\*\*\*\*\*\*\*\*\*\*\*\*\*\*\*\*\*\*\*\*\*\*\*\*\*\*\*\*\*\*\*\*\*\*\*\*\*\*\*\*\*\*\*\*

Figure 11: The figure panel corresponding to Forced_slab (Figure 11a) is missing

\*\*\*\*\*\*\*\*\*\*\*\*\*\*\*\*\*\*\*\*\*\*\*\*\*\*\*\*\*\*\*\*\*\*\*\*\*\*\*\*\*\*\*\*\*\*\*\*\*\*\*\*\*\*\*\*\*\*\*\*\*\*\*\*\*\*\*\*\*\*\*\*\*\*\*\*\*\*\*\* No panel is missing, as there is no thermocline anomaly in the case of slab ocean model. However, the figure caption was incorrectly referring to 2 panels. The caption has been corrected now. \*\*\*\*\*\*\*\*\*\*\*\*\*\*\*\*\*\*\*\*\*\*\*\*\*\*\*\*\*\*\*\*\*\*\*\*\*\*\*\*\*\*\*\*\*\*\*\*\*\*\*\*\*\*\*\*\*\*\*\*\*\*\*\*\*\*\*\*\*\*\*\*\*\*\*\*\*

Please also note the supplement to this comment:
https://www.earth-syst-dynam-discuss.net/esd-2017-113/esd-2017-113-AC2-supplement.pdf